# Molecular Iodine Supplement Prevents Streptozotocin-Induced Pancreatic Alterations in Mice

**DOI:** 10.3390/nu14030715

**Published:** 2022-02-08

**Authors:** Julia Rodríguez-Castelán, Evangelina Delgado-González, Valentin Varela-Floriano, Brenda Anguiano, Carmen Aceves

**Affiliations:** Instituto de Neurobiología, Universidad Nacional Autónoma de México, Juriquilla, Querétaro 76230, Mexico; julie.rocas@hotmail.com (J.R.-C.); edelgado@comunidad.unam.mx (E.D.-G.); varvale1402@outlook.com (V.V.-F.); anguianoo@unam.mx (B.A.)

**Keywords:** molecular iodine, pancreatitis, streptozotocin, PPARγ, Nrf2

## Abstract

Pancreatitis has been implicated in the development and progression of type 2 diabetes and cancer. The pancreas uptakes molecular iodine (I_2_), which has anti-inflammatory and antioxidant effects. The present work analyzes whether oral I_2_ supplementation prevents the pancreatic alterations promoted by low doses of streptozotocin (STZ). CD1 mice (12 weeks old) were divided into the following groups: control; STZ (20 mg/kg/day, i.p. for five days); I_2_ (0.2 mg/Kg/day in drinking water for 15 days); and combined (STZ + I_2_). Inflammation (Masson’s trichrome and periodic acid–Schiff stain), hyperglycemia, decreased β-cells and increased α-cells in pancreas were observed in male and female animals with STZ. These animals also showed pancreatic increases in immune cells and inflammation markers as tumor necrosis factor-alpha, transforming growth factor-beta and inducible nitric oxide synthase with a higher amount of activated pancreatic stellate cells (PSCs). The I_2_ supplement prevented the harmful effect of STZ, maintaining normal pancreatic morphometry and functions. The elevation of the nuclear factor erythroid-2 (Nrf2) and peroxisome proliferator-activated receptor type gamma (PPARγ) contents was associated with the preservation of normal glycemia and lipoperoxidation. In conclusion, a moderated supplement of I_2_ prevents the deleterious effects of STZ in the pancreas, possibly through antioxidant and antifibrotic mechanisms including Nrf2 and PPARγ activation.

## 1. Introduction

Acute and chronic pancreatitis are considered the inflammatory processes involved in the development and progression of type 2 diabetes mellitus and pancreatic ductal adenocarcinoma [1]. Although its pathogenesis is not fully described, oxidative stress is clearly involved in its initiation. During acute pancreatitis, there is a rapid generation of reactive oxygen species (ROS) and calcium storage that cause damage to the pancreatic cells [2]. Chronic pancreatitis appears to be the result of recurrent injury and defective repair, leading to generalized pancreatic atrophy and fibrosis [3]. Pancreatic stellate cells (PSCs) are crucial in the installation and progression of these processes [4]. The most common marker of acute pancreatitis is the transformation of quiescent PSCs (positive to glial fibrillary acidic protein (GFAP)) into activated PSCs (positive to α-smooth muscle actin (α-SMA)), and tumor necrosis factor alpha (TNFα) seems to be mainly responsible for its activation [4]. TNFα is an inflammatory cytokine produced by macrophages/monocytes during acute inflammation [5]. Activated PSCs promote β-cell apoptosis or transdifferentiation into α-cells, generating glucose intolerance [6] and they exhibit a high capacity to express growth factors (transforming growth factor-beta (TGFβ) and vascular endothelial growth factor (VEGF)), proinflammatory cytokines (TNFα, IL6, IL8 and IL10) and autocrine factors (cyclooxygenase-2 (Cox2), Collagen 1, activin A and inducible nitric oxide synthase (iNOS)) [3,5]. Natural compounds with antioxidant properties, such as resveratrol or curcumin, inhibit the PSCs activation, decreasing the production of ROS and collagen in cell cultures [7,8,9] and in vivo mouse models [10]; however, at the clinical level, antioxidant treatments with Beta-carotene, N-acetyl-L-cysteine (NAC), vitamin E and selenium have not been shown to be sufficient to reduce acute pancreatitis, which indicates a greater complexity in the development of these inflammatory processes [11,12].

Several studies have shown that molecular iodine (I_2_) is a powerful antioxidant [13]. Its reductive capacity in vitro (by the ferric reducing antioxidant power assay) is ten times more efficient than that of ascorbic acid [14]; whereas in vivo, I_2_ at micromolar amounts increases the antioxidant status in several tissues and species [15,16,17]. I_2_ also induces the activation of the nuclear factor erythroid-2 (Nrf2) pathway, triggering the expression of several phase II protective antioxidant enzymes such as catalase (Cat) and superoxide dismutase type 1 (Sod1) [18]. Furthermore, I_2_ binds to arachidonic acid and generates 6-iodolactone (6-IL), which activates peroxisome proliferator-activated receptor type gamma (PPARγ), exerting metabolic, antioxidant and immunoregulatory effects [19,20]. The actions of I_2_ in the pancreas are not well studied, but moderate I_2_ supplementation prevents the incidence rate of insulitis in the spontaneous diabetes mellitus bio-breeding/Worcester rat model (BB rats) [21] and reduces the oxidative pancreatic damage secondary to hypothyroidism [22]. The purpose of the present study was to evaluate oral I_2_ supplementation in the initiation and progression of pancreatitis using a prediabetic mouse model generated by low doses of streptozotocin (STZ).

## 2. Materials and Methods

### 2.1. Reagents

Food (Purina certified rodent chow, Ralston Purina Co., St. Louis, MO, USA), iodine (Macron Avantor, Radnor, PA, USA), streptozotocin (Sigma-Aldrich, St. Louis, MO, USA) were used.

### 2.2. Animals

Twelve-week-old CD1 male mice were housed under controlled temperature (22 ± 1 °C) and a light:dark cycle of 12:12 h. They were provided with pellet food and water ad libitum; the total consumption was collected daily. The handling and sacrifice of the animals was reviewed and approved by the Research Ethics Committee of the Institute of Neurobiology (INB-UNAM; Protocol 35), which is supported by the Animal Care and Use Program of the National Institutes of Health (NIH, Bethesda, MD, USA). Mice were randomly assigned to four experimental groups (*n* = 6): control (C); streptozotocin (STZ, 20 mg/Kg/day); I_2_ (oral 0.2 mg/Kg/day of I_2_ in drinking water); and STZ + I_2_ (STZ + I_2_). Groups with I_2_ were supplemented for the 15 days of treatment. STZ groups were intraperitoneally injected for 5 consecutive days from day 8, at 14:00 h. The STZ was solubilized in phosphate-buffered saline (PBS; pH 7.4) immediately before use within 5 min of preparation [22]. Blood glucose was measured from the tail vein using Accu-Chek (Active, Roche, Germany), at the start of treatment (day 0), before the first dose of STZ (day 8), after 3 injections of STZ (day 11), and at the end of treatment (day 15). The animals were fasted for 6 h (8:00–14:00 h) on the days of measurement. On day 15, mice were euthanized under anesthesia with a mixture of ketamine/xylazine (30 and 6 mg/kg body weight). Immediately after death, a part of the pancreas was collected, histologically processed, embedded in Paraplast X-TRA (Leica, Weztlar, Germany) and longitudinally cut at a thickness of 4 μm using a microtome (Leica RM2125 RTS, Weztlar, Germany). Another part of the pancreas was frozen at −70 °C for biochemical measures. To evaluate the possible differential response of the two sexes, sixteen CD1 female mice at 12 weeks old were randomly divided into C, STZ, I_2_ and STZ + I_2_ groups (*n* = 4) under the same conditions as males and followed for 22 days.

### 2.3. Pancreatitis and Insulitis Determination

Pancreas samples were stained with hematoxylin/eosin and periodic acid–Schiff (PAS) to evaluate the severity of pancreatitis. Eight fields were randomly chosen to determine different parameters. A semiquantitative assessment was carried out by two expert viewers analyzed as follows: edema (elongated white spaces between pancreatic lobules or between acinar cells); acinar necrosis (loss of acinar cells structure with pyknotic nuclei); inflammatory cell infiltrate (immune cells inside acinar space); and intracellular vacuolization (formation of a vesicle with circular shape inside acinar cells) according to previous reports [23]. The grades of pancreatitis were measured for each parameter considering: 0 (absence, or less than 10%); 1 (low presence, or less than 25%); 2 (moderate presence, or between 25% and 50%); and 3 (high presence, or higher than 50%). The grades of insulitis were evaluated as follows: without insulitis (score 0); peri-insulitis (immune infiltration around islets, score 1); intermediate insulitis (<30% immune infiltration, score 2); severe insulitis (<50%, score 3); and destructive insulitis (>50% immune infiltration, score 4) according to previous reports [24,25]. Overall, islets were analyzed and classified depending on the grade of insulitis mentioned above, given a score between 0 and 4. Data were collected from 30 to 50 islets from the pancreas of each mouse (six in every treatment). Masson’s trichrome was used to identify the presence of collagen (associated with fibrosis).

### 2.4. Morphology, Vasculature and iNOS Presence

Morphometric analysis and immunohistochemistry for GFAP, α-SMA, insulin, glucagon, vasculature (CD34) and iNOS were performed using an optical microscope at 40× (Leica DM250, Wetzlar, Germany). Slides of pancreas samples (4 μm) were deparaffinized and incubated in 10 mM sodium citrate pH 6 at 80 °C for 30 min to retrieve antigens. Endogenous peroxidases were quenched with 0.3% hydrogen peroxide diluted in PBS. Endogenous binding sites for secondary antibodies were blocked with 20% bovine serum albumin (BSA) diluted in PBS with 0.3% Triton X-100 (PBST) for 1 h. Independent sections were incubated with antibodies (Appendix A). Subsequently, they were incubated with biotinylated secondary antibody (Appendix A) diluted in PBST for 2 h at 37 °C. Immunostaining was developed according to the Vectastain ABC kit directions (Vector Labs, Burlingame, CA, USA). Sections were counterstained with McGill’s hematoxylin. Non-specific immunostaining was observed when primary antiserum was omitted. Six islet fields were randomly chosen and photographed at 40× or 100× with a digital camera (Leica DFC420, Wetzlar, Germany) to quantify the positive stain per field.

### 2.5. Gene Expression

Quantitative real-time RT-PCR (RT qPCR) was used to analyze the pancreatic expression of Sod1, and Cat as previously described [20]. Briefly, the total RNA was obtained using Trizol reagent (Life Technologies, Inc., Carlsbad, CA, USA). Two micrograms of RNA were reverse transcribed (RT) using oligo-deoxythymidine (Invitrogen, Waltham, MA, USA). DNA amplification was performed with Maxima SYBR Green (Thermo Fisher, Waltham, MA, USA) on the Rotor-Gene 3000 sequence detector system (Corbett Research, Mortlake, NSW, Australia). The relative levels of mRNA were normalized to the level of β-actin mRNA. cDNA amplification was carried out with the primers described in Appendix A.

### 2.6. Protein Extraction and Western Blotting

TGFβ, VEGF, TNFα, IL10, CD8a, Nrf2 and PPARγ (wild-type and 112-phosphorylated) were quantified. Pancreas tissue (~50 mg) was disrupted using an electronic homogenizer in RIPA buffer added with protease inhibitor cocktail (mini-Complete EDTA inhibitor; Roche Diagnostics GmbH, Rotkreuz, Switzerland). Protein extracts (50 µg) were denatured in Laemmli sample buffer, resolved in 10% or 15% SDS-PAGE, and electro-blotted to nitrocellulose membranes (Bio-Rad, Hercules, CA, USA). To confirm that protein content was equal in all lines, membranes were stained with Ponceau’s Red. Membranes were soaked with 5.0% non-fat dry milk (Bio-Rad, Hercules, CA, USA), diluted in PBS containing 0.1% tween-20 (PBST) and incubated overnight at 4 °C with antibody (Appendix A). Later, membranes were incubated with a secondary antibody (Appendix A) conjugated with horseradish peroxidase at room temperature under constant agitation for 1 h. Immunoreactivity was enhanced by chemiluminescent substrate (Bio-Rad, Hercules, CA, USA) according to the manufacturer’s instructions, followed by exposure to radiographic film (Amersham, Biosciences, Little Chalfont, UK). To correct the differences in the total protein loaded in each lane, the protein content was normalized using Actin as an internal control. Blots were stripped with a 0.1 M glycine solution (pH 2.5, 0.5% SDS) for 2 h at 37 °C and incubated with antibody (Appendix A) at 4 °C overnight. Blots were incubated with a secondary antibody (Santa Cruz Biotechnology, Dallas, TX, USA) at room temperature for 1 h under constant agitation.

### 2.7. Lipid Peroxidation in Pancreas

The pancreas sample (25 mg) was homogenized in ice-cold tris buffer (20 mM, pH 7.4) and centrifuged at 3000 rpm for 10 min at 4 °C. Supernatant was collected and immediately tested with the lipid peroxidation microplate assay (Oxford Medical Research, Inc., St. Louis, MO, USA). The kit used the thiobarbituric acid reaction, and lipoperoxidation is expressed as micromoles of malondialdehyde (MDA) per micrograms of protein of pancreas.

### 2.8. Statistical Analyses

Statistical analyses were performed with GraphPad Prism v6.01 (GraphPad Software, La Jolla, San Diego, CA, USA). Results were expressed as mean ± SEM. One-way ANOVA and Tukey’s post hoc test were used to determine the significant differences between all groups (*p* < 0.05). In the figures, different letters indicate statistical differences between groups.

## 3. Results

Iodine supplement prevents the diabetes profile induced by STZ.

Figure 1 summarizes the body weight gain, food and water consumption, as well as circulating levels of glucose in male and female mice. In both sexes, the reduction in body weight gain and increase in water and food consumption within the second week of STZ administration confirms the effect of the drug. The increase in glucose, indicating diabetes installment, was observed from day 11 in males and day 15 day in females. The I_2_ supplement in control animals does not generate any modification in these parameters. The coadministration of both components (STZ + I_2_ group) prevented the increase in food and water consumption, although the increase in glucose levels was only partially prevented, and the weight loss associated with STZ remained in males and females until day 15. However, glucose and body weight in STZ + I_2_ females were totally recovered by day 21. All these data suggest that the installment of the diabetes profile and its prevention by I_2_ are similar in both sexes. All other parameters were only analyzed in the male pancreas.

Pancreatitis, destructive islets and hypervascularization were not present in I_2_-supplemented animals.

Figure 2 describes the pancreatic histological score showing that STZ is accompanied by pancreatitis with significant increases in edema, necrosis as well as inflammatory infiltration. Masson’s stains show increases in intrapancreatic fibrosis. These alterations were also accompanied by an increase in the number and cross-sectional area (CSA) of islets. The I_2_ supplement prevented the inflammation and maintained the normal number of islets, although the CSA remained increased.

The integrative analysis of insulitis is summarized in Figure 3. PAS and Masson´s stain show inflammation and necrotic areas. It is evident that STZ generated severe damage in islets marked by 45% of destructive and 30% of intermediate insulitis. Vascularization (CD34+ cells) was increased around the islets associated with an increase in VEGF expression. The iodine supplement exhibited significant protection showing mainly peri-insulitis (60%) and intermediate or severe conditions only in 15% and 10% of islets. The I_2_ supplement also prevented the vessel increase and VEGF induction.

β-cell transdifferentiation and PSC activation were prevented in animals with I_2_ supplementation.

Cellular analysis showed that STZ administration modified the presence and functionality of β- (insulin) and α- (glucagon) cells (Figure 4). STZ promoted a decrease in β-cell density and an increase in α-cell density in islets, suggesting an early diabetogenic effect (Figure 4a). GFAP and α-SMA immunohistochemistry were used to evaluate the quiescent or activated PSCs, respectively (Figure 4b). Quiescent PSCs decreased, whereas activated PSCs increased in the pancreas of STZ animals. The I_2_ supplement prevented changes in cell proportion and showed a partial but significant protection against PSC activation.

Figure 5 shows the markers associated with the inflammatory immune response. STZ administration increased the presence of TNFα, IL10, TGFβ and CD8a in pancreas tissue (Figure 5a). iNOS was highly expressed in the acinar and islet cells of STZ animals (Figure 5b). The I_2_ supplement attenuated these proinflammatory conditions by impairing the activation intensity of the immune response.

PPARγ and antioxidants-related gene expression in the pancreatic tissue.

Figure 6 summarizes the proposed explanation for the preventive action of I_2_ supplementation in the STZ model. The higher amount of MDA in STZ mice corroborates the pancreatic oxidative state of these animals. The partial increase in antioxidant markers, such as Nrf2, Sod1 and Cat, indicates a physiological response to these conditions, which, however, seems insufficient to neutralize lipoperoxidation. I_2_ supplementation exerted an effective antioxidant effect by eliciting the highest expression of antioxidant markers, thus preventing the oxidative environment (MDA at basal levels). The presence of the I_2_ supplement also prevented the 112 phosphorylation of PPARγ, maintaining its activated form.

## 4. Discussion

STZ induces an uncontrolled inflammatory response in β-cells that develops into the destruction of these cells; however, continuous applications at low concentration (20 mg/Kg/day for 5 days) could be used as a prediabetic model [26,27]. We used this model to analyze the effect of I_2_ in the prevention of pancreatic injury. STZ administration was accompanied by polyphagia, polydipsia, hyperglycemia and reduced body weight after the second week (days 8–15). These characteristics were observed in both sexes, although a significant increase in glucose occurred in males starting on day 11 and in females until day 15. Several studies have indicated that women are less susceptible to diabetes because 17β-estradiol protects pancreatic β-cells from oxidative injury [28]. However, recent metanalyses have shown that this protection is more complex and could be determined by sex differences (i.e., sex chromosomes, sex-specific gene expression and sex hormones) and gender differences (e.g., differences in lifestyle, environmental influences and nutrition) [29]. Reports in animals have also shown clear estrogen association [30] or only transient prevention [31]. Our study corroborated that in mice, STZ treatment exhibited a transient prevention of one week in females, but after 22 days both sexes exhibited the same pre-diabetic pattern. The significant decrease in body weight might have resulted from severe polyuria, which accompanied the untreated hyperglycemia [27]. We observed that STZ and I_2_ diminished polyphagia and polydipsia but only partially prevented hyperglycemia and body weight loss during the first two weeks. However, in the third week (day 22) of I_2_ treatment, body weight gain and normoglycemia were observed in female animals, suggesting a sustained beneficial effect of the I_2_ supplement on both sexes. These results agree with previous reports showing that the moderate supplementation of I_2_ prevents the incidence of insulitis in BB rats [21] and that I_2_ has a preventive effect on the oxidative pancreatic damage associated with hypothyroidism [22]. At the pancreatic level, STZ animals showed edema, necrosis and inflammatory infiltration accompanied by severe damage in islets marked by 45% of destructive and 30% of intermediate insulitis and a compensatory response increasing angiogenesis around the islets associated with increases in VEGF expression. Additionally, this group exhibited significant functional pancreatic damage, changing the density proportion between β- and α-cells, as well as clear PSCs activation. The toxic effect of STZ has been explained by its selective uptake into β-cells via the low-affinity glucose transporter (GLUT2) present in their plasma membrane, generating oxidative stress, which damages the cell structure, and insulin secretion, as well as a proinflammatory environmental inductor of PSCs activation [26]. Active PSCs have been associated with fibrosis because they can be transformed into myofibroblast-like cells that are the main source of the extracellular matrix [4]. In our study, the I_2_ supplement decreased the positive areas of α-SMA–stained sections, diminished the content of the primary PSCs activators TNFα and TGFβ, and exerted a protective effect on β-cell integrity. The molecular mechanisms of I_2_ that inhibit pancreatic damage can be explained by complementary actions: first, by the great reducing capacity of iodine, which is 10 times greater than that of ascorbic acid and 60 times greater than that of potassium iodide [13] and could be related to the significant decrease in MDA levels observed in the present work. Second, through the release/activation of Nrf2. Recently, Nrf2 has emerged as an indispensable regulator of the inducible cytoprotective genes in various tissues including the pancreas [32]. In response to oxidative stress, Nrf2 accumulates in the nucleus, where it binds to an antioxidant response element (ARE) in the regulatory sequences of its target genes which encode antioxidant enzymes and de-toxify proteins [33]. Under basal conditions, Nrf2 is anchored to the cytoplasm through the actin cytoskeleton-binding protein 1 (KEAP1). The iodination of KEAP1 results in the release and translocation of Nrf2 to the nucleus and the increase in Sod1, NAD(P)H quinone dehydrogenase 1 (NQO1) and glutathione S-transferase P1 (GSTP1) expression [17]. Our data agree with the increase in Nrf2 and the overexpression of Sod1 and Cat enzymes. Although these results support that the antioxidant effect is important, clinical studies have shown that therapies with only antioxidants (Beta-carotene and vitamin E) are not sufficient and may even increase mortality in several diseases [11,12]. Therefore, we consider that the third plausible mechanism involved in the beneficial effect of I_2_ is the activation of the nuclear receptor PPARγ. The STZ group principally showed that p-PPARγ, an inactive form of these receptors, whereas the active form (wild-type PPARγ) increased in the I_2_ and STZ + I_2_ groups. Growth factors, such as TGFβ and platelet-derived growth factor, were shown to phosphorylate PPARγ via the MAP kinase signaling pathway and decrease PPAR transcriptional activity [34]. The prevalence of wild-type PPARγ (active) observed in the STZ + I_2_ group agrees with previous reports showing that I_2_, through its conversion to 6-IL, is a specific agonist of PPARγ [18]. The expression of PPARγ is negatively correlated with PSC proliferation, and treatment with the agonist troglitazone reduces α-SMA expression and PSCs’ proliferation, making the activated PSCs transform into the quiescent state [35] and preventing fibrosis development [36]. Studies are required to specifically analyze the effect of iodine on PSCs. Additionally, the therapeutic potential that the I_2_ supplement may have in metabolic syndrome or even in well-established diabetes is a goal in our perspective.

## 5. Conclusions

Molecular iodine supplement prevents hyperglycemia, pancreatitis, insulitis and the inflammatory effects of moderate doses of STZ in the short term in male and female mice. Antioxidant activity and PPARγ activation are proposed as mechanisms in these preventive effects. Further studies are necessary to elucidate the potentially therapeutic effect of I_2_ in the management of several inflammatory diseases in the pancreas.

## Figures and Tables

**Figure 1 nutrients-14-00715-f001:**
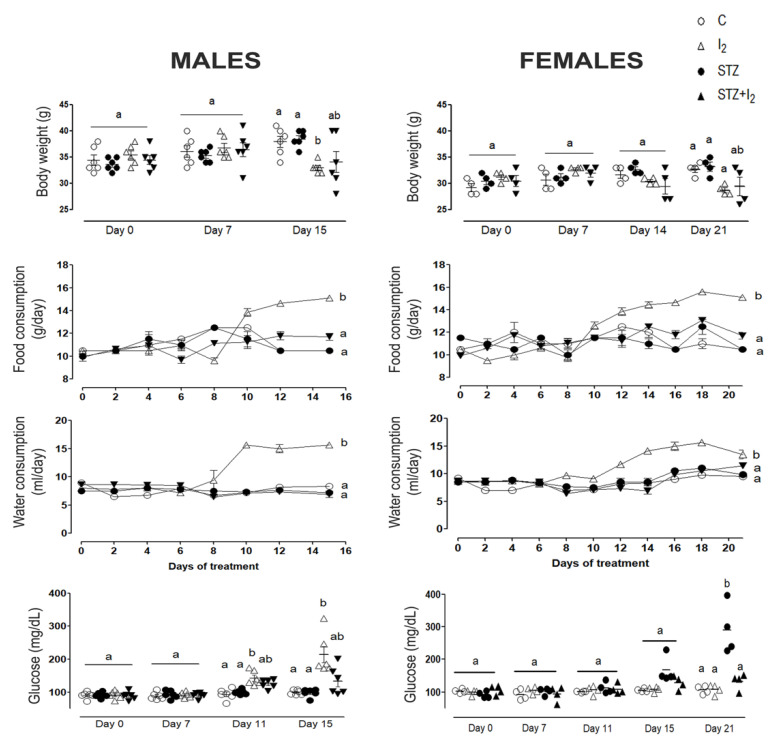
Body weight change, food and water consumption and glucose levels in male and female rats. Results are expressed as mean ± SEM. Different letters indicate statistical differences between groups (one-way ANOVA, Tukey’s test; *p* < 0.05).

**Figure 2 nutrients-14-00715-f002:**
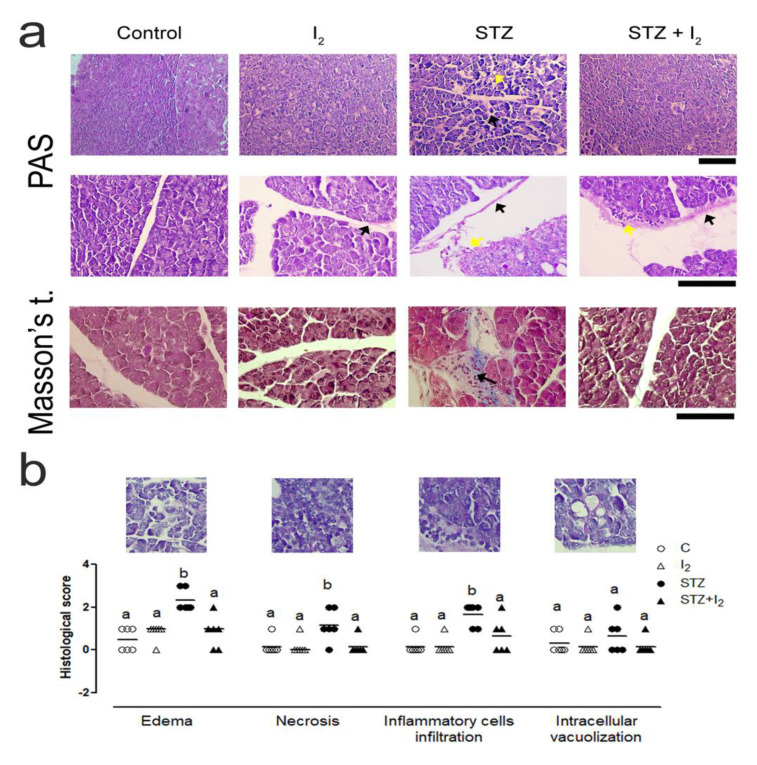
Histological characteristics of pancreatic acinus: (**a**) micrographs showing the presence of proteoglycans (black arrows, in Periodic Acid-Schiff stain), edema (black arrows, in PAS stain) and inflammatory cell infiltration (yellow arrows, in PAS stain) and collagen (black arrows in Masson’s trichrome). Scale: PAS stain 100 µm (first row); and 20 µm (second row); Masson’s t. 20 µm; and (**b**) representative micrographs of pancreas damage (20 µm) and quantitative analysis of the histological score. Data are expressed as the mean ± SEM, and different letters indicate a statistical difference between groups (one-way ANOVA, Tukey’s test; *p* < 0.05).

**Figure 3 nutrients-14-00715-f003:**
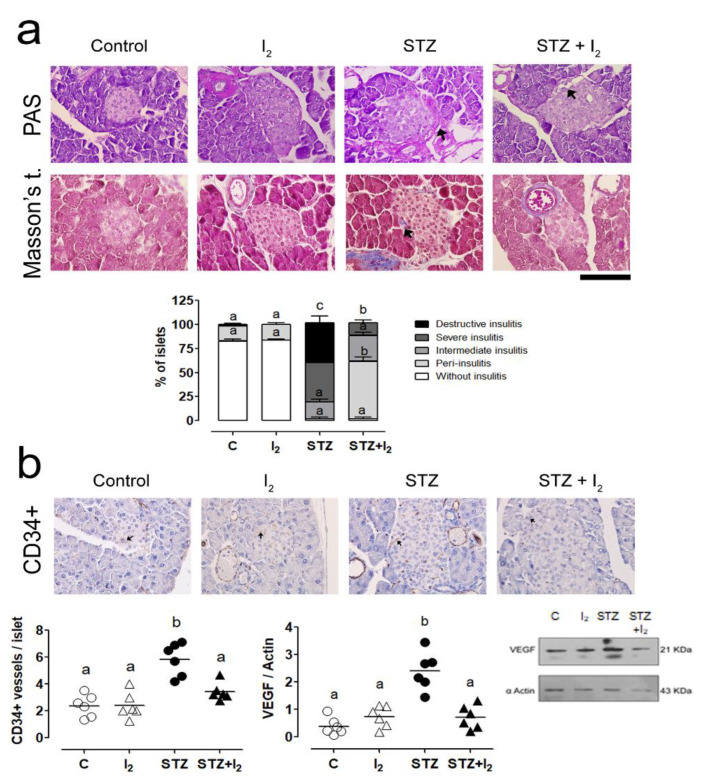
Morphological pancreatic islets and vasculature: (**a**) PAS stain micrography showing proteoglycans and immune cell infiltration (black arrows, in PAS stain) and collagen (black arrows, in Masson’s stain) in pancreatic islets. Quantitative analysis (% of islet) of the degree of insulitis according to the histological characteristics; (**b**) microphotography of vessels (CD34+ protein) and quantification (Western blot); and VEGF quantification (Western blot). Scale = 50 µm. Data are expressed as mean ± SEM, and different letters indicate a statistical difference between groups (one-way ANOVA, Tukey’s test; *p* < 0.05), STZ, streptozotocin.

**Figure 4 nutrients-14-00715-f004:**
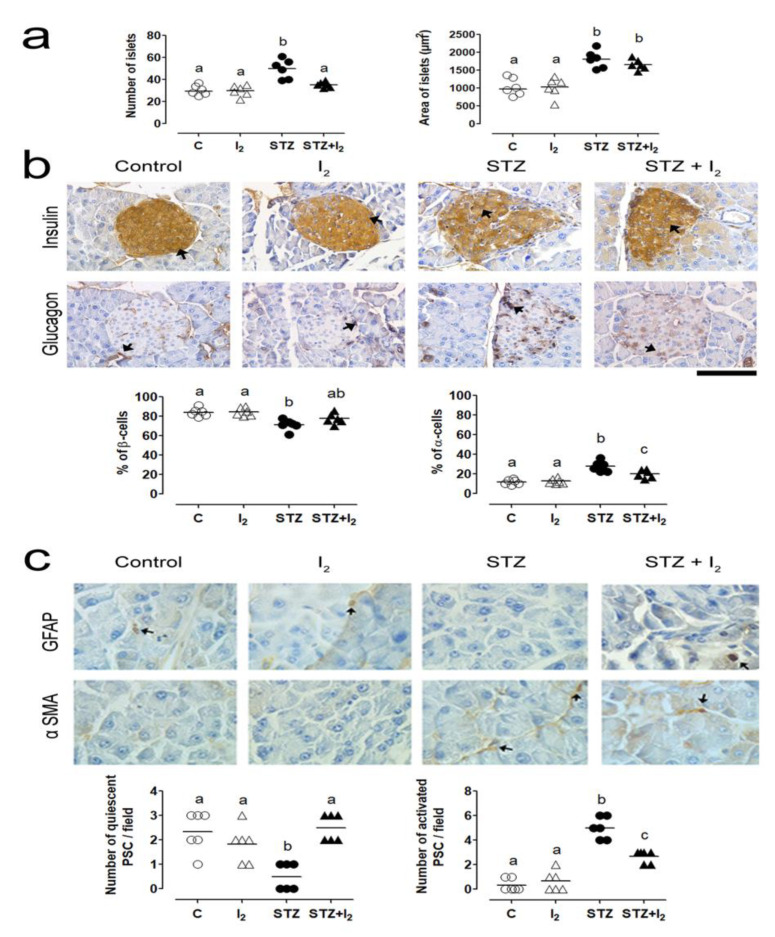
Localization and quantification of α- and β-cells and Pancreatic stellate cells (PSC )activation. The characteristics of pancreatic islets: (**a**) the number and area of islets; (**b**) the microphotography of insulin and glucagon immune stain (50 µm, black arrows) and α-cells and β-cells quantification (%); and (**c**) microphotography (20 µm) of quiescent PSCs (glial fibrillary acidic protein (GFAP)+, immunohistochemistry, black arrows), active PSCs (α-smooth muscle actin α-SMA, black arrows) and quantifications per field. Data are expressed as the mean ± SEM, and different letters indicate a statistical difference between groups (one-way ANOVA, Tukey’s test; *p* < 0.05). Pancreatic stellate cells (PCS), glial fibrillary acidic protein (GFAP), α-smooth muscle actin (α-SMA).

**Figure 5 nutrients-14-00715-f005:**
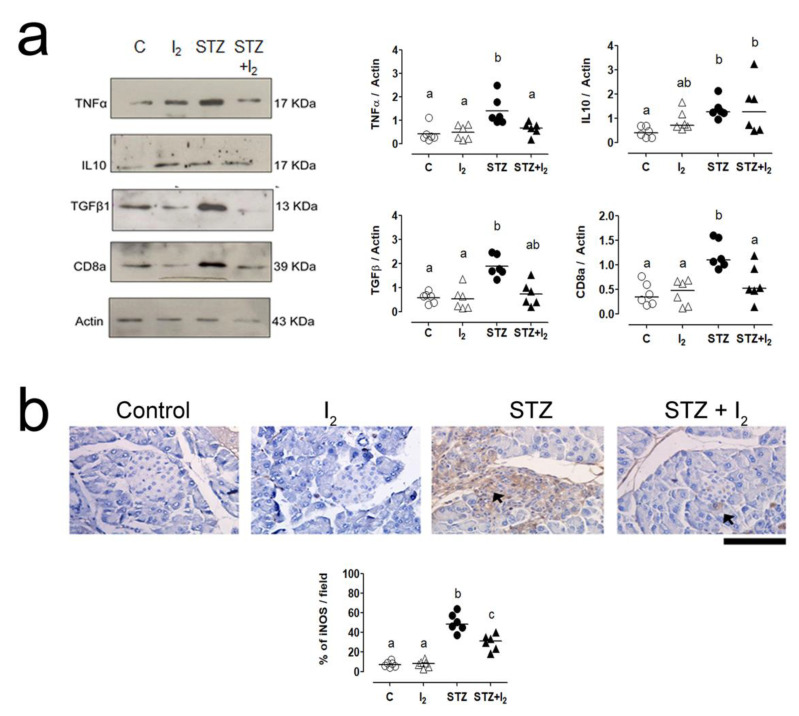
Inflammation markers in pancreas: (**a**) tumor necrosis factor alpha (TNFα), IL10, transforming growth factor-beta (TGFβ) and CD8-a protein (CD8-a) quantification; (**b**) microphotography of representative iNOS (50 µm, black arrows) and quantification. Data are expressed as mean ± SEM, and different letters indicate a statistical difference between groups (one-way ANOVA, Tukey’s test; *p* < 0.05).

**Figure 6 nutrients-14-00715-f006:**
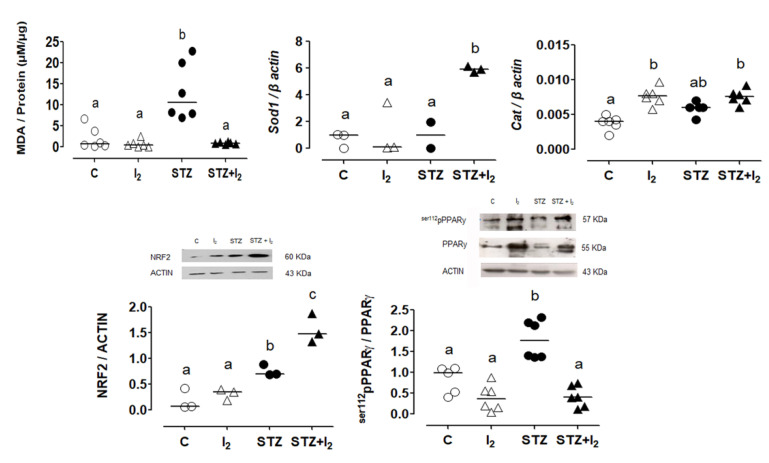
Pancreatic antioxidant status and peroxisome proliferator-activated receptor type gamma PPARγ phosphorylation. Lipoperoxidation (expressed as micromoles of malondialdehyde (MDA) per micrograms of protein), superoxide dismutase type 1 (Sod1) and catalase (Cat) expression (qRT-PCR). Amount (Western blot) of NF-E2-related factor 2 (Nrf2) and PPARγ. The deactivated PPARγ amount was calculated by the phosphorylated PPARγ/active PPARγ ratio. Data are expressed as the mean ± SEM, and different letters indicate a statistical difference between groups (one-way ANOVA, Tukey’s test; *p* < 0.05). PPAR: peroxisome proliferator-activated receptor type gamma.

## Data Availability

The data that support the findings of this study are available from the corresponding author upon reasonable request.

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
