# Peer review of "Molecular Iodine Supplement Prevents Streptozotocin-Induced Pancreatic Alterations in Mice"

_nutrients, 2022, doi:10.3390/nu14030715_

Round 1

Reviewer 1 Report

My comments have been largely addressed.

Author Response

My comments have been largely addressed.

We offer an apology to the reviewer. We carefully analyzed his observations and corrected figure 1, removing the letters that divided the figure (they were not necessary), we enlarged the labels and numbering of all the figures. we include labels on all micrographs and western blots. We describe the results in more detail.

 We appreciate your comments, which undoubtedly improved our manuscript

Reviewer 2 Report

The paper describes the characterization of molecular Iodine treatment as a potential approach to treat STZ-induced pancreatic alterations. Authors propose that Iodium elicits pancreatic pro-survival effects potentially by modulating NRF2 and PPARy signalling networks. The manuscript addresses an important question (how to improve treatment of pancreatic diseases), highlights a novel approach, and shows its initial potential.

The authors have now slightly improved the Figures presentation but it still below the average. E.G. Figure 1B-C-D are still very difficult to interpret. Western blot in Figures 3B and 5A are missing the labels.

Authors also suggest that the third plausible mechanism is trough pPPARγ based on Figure 6C. There is no loading control in this western blot and it is not clear if PPARγ wild and phosphorylated form were detected with the same antibody or by using two different antibodies against the two isoforms. It is not clear by this image if there is a difference between Control and STZ samples. Are also the densitometry analyses throughout the manuscript performed on six different western blots?  

Authors did not state how many mice were used for the study. Is 6 mice per group? 

Author Response

The authors have now slightly improved the Figures presentation but it still below the average. E.G. Figure 1B-C-D are still very difficult to interpret. Western blot in figures 3B and 5A are missing labels.

 The referee is right. We carefully analyzed the manuscript and the figures. We corrected errors (letters not necessary, figure 1 A-D), we added identification paragraphs in the western blots and in the micrographs.

Authors also suggest that the third plausible mechanism is through pPPARγ based on figure 6C. There is no loading control in this western blot and it is not clear if PPARγ wild and phosphorylated form were detected with the same antibody or by using two different antibodies against the two isoforms. It is not clear by this image if there is a difference between Control and SZT samples. Are also the densimetry analyses throughout the manuscript performed on six different westerns blots?

We analyzed the amount of PPARγ and phosphorylated PPARγ (pPPARγ) with specific antibodies for each protein (see table S2). The antibody to PPARγ sees only the native protein (the active receptor), whereas the antibody to pPPARγ labels both the native (55 KD) and the phosphorylated (57 KD) proteins as they appeared in western blot photographs. To quantify which receptor was preponderant in each group, we divided the quantification of the phosphorylated (inactive) by the native (active) pPPARγ /PPARγ ratio. Our results are the result of 6 individual westerns for each protein and for each group.

 We include part of this explanation in the figure caption

Author did not state how many mice were used for the study. Is 6 mice per group?

The referee is right; We used six male mice for each treatment and three female mice per group. These clarifications are included in the materials section.

 We appreciate your comments and hope we have improved the work.

Reviewer 3 Report

Thank you for the oppurtunity of reviweing this paper,

Overall, this is an interesting paper based on heavilty chemical and immunological analysis, that merits to be considered for acceptance

I have some comments,

  • How the authors decided the percent of the tissue destruction as it written "It is evident that STZ generated severe dam- 194
    age in islets marked by 45% of destructive and 30% of intermediate insulitis" in line 194 and so on ??
  • The quality of the figures is not good, and the words and the legends within the figure are rather small which make very difficult for the readers, and these should be corrected.
  • What the authors meant by ". Different letters indicate statistical differences between groups." in the statistical analysis section
  • Why the Mice were randomly assigned to four experimental groups, what rendomization was used in this study ?? does all mice are at the same age and healthy ??
  • Please add reference to "During acute pancreatitis, there is a rapid generation of reactive oxygen species (ROS) that cause damage on the pancreatic cells" in the introduction section
  • Please add reference to "Pancreatic stellate cells (PSCs) are crucial in the installation and progression of these processes" in the introduction section.
  • Please state which antioxidants treatment were used "however, at the clinical level, antioxidant treatments have not been shown to be sufficient to reduce acute pancreatitis" in the introduction section

Author Response

How the authors decided the percent of the tissued destruction as it written “It is evident that STZ generated severe damage in islets marked by 45% of destructive and 30% of intermediate insulitis” in line 194 and so on?

We analyzed around 30-50 islets per animal in each treatment. We classified islets by degree of insulitis, using a previously reported score that includes: without insulitis (score 0), peri-insulitis (immune infiltration around islets, score 1), in-intermediate insulitis (<30% immune infiltration, score 2) , severe insulitis (<50%, score 3), and destructive insulitis (>50% immune infiltration, score 4). The detailed description of this quantification is included in the methodology section.

The quality of the figures is not good, and the words and the legends within the figure are rather small which make very difficult for the readers, and these should be corrected.

We thank the observation, the quality of figures was improved, the legends of figures were modified.

What the authors meant by “Different letters indicate statistical differences between groups” in the statistical analysis section.

As mentioned in the description of the statistical analyses, one-way ANOVA and Tukey's post hoc test were used to determine significant differences between all groups at p<0.05. Letter notation is widely used in statistics. In the materials and methods section, we corrected the paragraph. “In the figures, different letters indicate statistical differences between groups.”.

Why the mice were randomly assigned to four experimental groups, what randomization was used in this study?? Does all mice are the same age and healthy??

In general, the mice had similar weight, normal blood levels and were healthy overall. Groups were randomly formed to combine animals from different litters.

 Please add reference to “During acute pancreatitis, there is a rapid generation of reactive oxygen species (ROS) that cause damage on the pancreatic cells” in the introduction section.

We thank the observation; we added the reference in introduction.

Please state which antioxidants treatment were used “however, at the clinical level, antioxidant treatments have not been shown to be sufficient to reduce acute pancreatitis” in the introduction section.

We thank the observation, the antioxidant treatment used in these studies were selenium, beta carotene, N-acetyl-L-cysteine (NAC), and vitamin E, we added them in the corresponding paragraph

We appreciate your comments and hope we have improved the work.

Round 2

Reviewer 2 Report

Authors have now addressed my comments

This manuscript is a resubmission of an earlier submission. The following is a list of the peer review reports and author responses from that submission.

Round 1

Reviewer 1 Report

Summary of work and major conclusions

The paper describes the characterisation of molecular Iodine treatment as a potential approach to treat STZ-induced pancreatic alterations. Authors propose that Iodium elicits pancreata pro-survival effects potentially by modulating NRF2 and PPARy signalling networks.  The manuscript addresses an important question (how to improve treatment of pancreatic diseases), highlights a novel approach and shows its initial potential.

General points: 

1. Quality of the figures is very poor. Authors should improved quality of the graphs, western blot and IHC images. It is quite impossible to draw any conclusions from these results. In Figure 1, I guess Iodine group was also swapped for STZ group according to what it is stated in the manuscript. Please also increase font size. There are not letters to describe the panels in the Figures 1, 2, 3 and 6 to help the reader. E.g. Figure 1A-B show..

2. Statistics. It is not very clear what 'a' or 'ab' stand for and this is not explained in the method. Authors should indicate statistical difference between groups showing the p value.    

Reviewer 2 Report

In my opinion, the manuscript is good overall, however it needs minor revision. More clear messages could be made in the Discussion and Introduction that the elementary iodine can reduce the damage to both endocrine and, secondarily, exocrine pancreas in the low-dose streptozotocin model of prediabetes. In the exocrine pancreas field the general consensus now is that added antioxidants generally do not protect against pancreatitis. This is because they tend to protect against apoptotic cell death first, thus paradoxically shifting the balance of the cell death from apoptosis to necrosis, leading to increased disease severity (https://pubmed.ncbi.nlm.nih.gov/21861696/). Having said that, the level of ROS (such as MDA) in the pancreas still correlates with the severity of pancreatitis. Thus, if the authors stipulate that I2 acts by being primarily an anti-oxidant, then the protective effect of I2 supplement on the exocrine pancreas is likely to be secondary to the effect on the endocrine pancreas. Perhaps this point needs to be reflected somewhere in the text. Please double check if the labels to Figure 1 are correct, that is if the hollow upward triangles correspond to I2 or STZ. Line34, replace "installation" with "initiation".